# Optimization of Laccase from *Ganoderma lucidum* Decolorizing Remazol Brilliant Blue R and *Glac1* as Main Laccase-Contributing Gene

**DOI:** 10.3390/molecules24213914

**Published:** 2019-10-30

**Authors:** Peng Qin, Yuetong Wu, Bilal Adil, Jie Wang, Yunfu Gu, Xiumei Yu, Ke Zhao, Xiaoping Zhang, Menggen Ma, Qiang Chen, Xiaoqiong Chen, Zongjin Zhang, Quanju Xiang

**Affiliations:** 1College of Resource, Sichuan Agricultural University, Chengdu 611130, China; otteyqin@163.com (P.Q.); wuyuetong666@163.com (Y.W.); bilal4jam@outlook.com (B.A.); 18875248226@163.com (J.W.); guyf@sicau.edu.cn (Y.G.); yuxiumeicool@163.com (X.Y.); zhaoke82@126.com (K.Z.); zhangxiaopingphd@126.com (X.Z.); mgen@sicau.edu.cn (M.M.); cqiang@sicau.edu.cn (Q.C.); 2Rice Research Institute of Sichuan Agricultural University, Chengdu 611130, China; xiaochenq777@126.com; 3Panzhihua Company of Sichuan Provincial Tobacco Corporation, Panzhihua 617026, China; ZhangzJ@163.com

**Keywords:** decolorization, laccase, *Ganoderma lucidum*, RT-PCR, isoenzyme genes

## Abstract

Many dyes and pigments are used in textile and printing industries, and their wastewater has been classed as a top source of pollution. Biodegradation of dyes by fungal laccase has great potential. In this work, the influence of reaction time, pH, temperature, dye concentration, metal ions, and mediators on laccase-catalyzed Remazol Brilliant Blue R dye (RBBR) decolorization were investigated in vitro using crude laccase from the white-rot fungus *Ganoderma lucidum*. The optimal decolorization percentage (50.3%) was achieved at 35 °C, pH 4.0, and 200 ppm RBBR in 30 min. The mediator effects from syringaldehyde, 1-hydroxybenzotriazole, and vanillin were compared, and 0.1 mM vanillin was found to obviously increase the decolorization percentage of RBBR to 98.7%. Laccase-mediated decolorization percentages significantly increased in the presence of 5 mM Na^+^ and Cu^2+^, and decolorization percentages reached 62.4% and 62.2%, respectively. Real-time fluorescence-quantitative PCR (RT-PCR) and protein mass spectrometry results showed that among the 15 laccase isoenzyme genes, *Glac1* was the main laccase-contributing gene, contributing the most to the laccase enzyme activity and decolorization process. These results also indicate that under optimal conditions, *G. lucidum* laccases, especially Glac1, have a strong potential to remove RBBR from reactive dye effluent.

## 1. Introduction

Over 10,000 different chemical dyes and dyeing auxiliaries are used worldwide in the textile and printing industries. Global dye production is estimated to be approximately 800,000 tons per annum, and at least 10% of used dye stuffs are discharged as environmental waste [1]. Due to their complex stable chemical structure and heavy metal content, dye effluent wastewater is relatively resistant to biodegradation. Presently, the removal of dye from wastewater depends mainly on physical or chemical processes, such as absorption, adsorption, coagulation–flocculation, ion exchange, and electrochemical methods. These methods are usually expensive and can produce toxic by-products, which can limit their application [2]. Biodegradation of dyes by microorganisms is an attractive method due to its low cost, high efficiency, absence of hazardous by-products, and environmental friendliness [3].

The application of white-rot fungi for the decolorization of industrial dyes, such as azo, heterocyclic, reactive, and polymeric dyes, has received increasing attention due to their ability to secrete non-specific oxidative enzymes, including lignin peroxidase, manganese peroxidase, and laccase [4,5,6]. Laccases (benzenediol: oxygen oxidoreductase, EC 1.10.3.2) are glycoproteins classified as low-specificity multi-copper oxidases, which can act on o- and *p*-quinols and catalyze one-electron oxidation of a wide range of organic and inorganic substrates by the reduction of molecular oxygen to water [7,8]. A typical laccase contains four copper ions per molecule, which are arranged into three distinct sites that perform different roles within a laccase domain [9]. Laccases are secreted extracellularly and have a high tolerance to high-concentration pollutants and low/high-pH environments. Therefore, they have attracted interest for detoxifying wastewater produced by pulp-bleaching processes, for industrial wastewater treatment, and for the decolorization of dye-containing effluents [10,11,12].

*Ganoderma lucidum* is a typical white-rot fungus, and has prominent biotechnological significance due to its medicinal and pharmacological properties. It also has great potential as a laccase producer for industrial dye decolorization [13,14], and plenty of studies have aimed to improve the yield of *G. lucidum* laccase [15]. Many reports have shown evidence that *G. lucidum* and its enzymes can be applied in the biodegradation of xenobiotics, including dyes [16], pesticide, and organic compounds [17,18]. Laccase was found to be a major enzyme produced by *G. lucidum* in most culture conditions [19,20], playing key roles in lignocellulose biomass hydrolysis [21] and has been used in various fields [22,23,24]. Sixteen laccase isoenzymes have been reported within laccase families from *G. lucidum* [25]. These isoenzymes often present differences in catalytic properties, regulation mechanisms, and localization [26].

The transcriptional profiles and enzymatic activities of laccases are often regulated by many environmental factors, including pH, temperature, substrate concentrations, metals, and mediators. The addition of RBBR inhibits the expression of the *Pleurotus ostreatus* laccase gene *pox1*, while Acetyl Yellow G (AYG) addition induces the highest expression of the *pox3* and *poxa1b* genes [27]. Metals can interact with the electron transport system of laccase and then affect its enzyme activity. The laccase from *G. lucidum* KMK2 was highly sensitive to Fe^2+^ and its enzyme activity was completely inhibited in the presence of Fe^2+^ [28]. Mediators can act as electron carriers between laccase and target compounds, making laccase able to oxidize not only phenolic compounds, but also non-phenolic compounds [29,30]. Adding mediators greatly affects the laccase stability, causes some loss of laccase activity, and this effect varies among different mediators [31]. The anthraquinone dye RBBR is one of the most important dyes used in the textile industry, and represents an important class of toxic and recalcitrant organo-pollutants. Various studies have been carried out to investigate RBBR decolorization [19,27,32]. High salt ion concentrations can significantly affect the role of *Ganoderma* laccase in decolorizing RBBR [33]. pH was also found to be an important parameter for enzymatic RBBR decolorization systems from *Lentinus crinitus* and *Psilocybecastanella* [34].

Although there is research reported for the decolorization of dyes by *G. lucidum* laccase, few studies have analyzed their effects on laccase expression. The main objective of this research was to investigate the optimal conditions for RBBR decolorization by *G. lucidum* laccase and analyze the main laccase-contributing genes among the 15 laccase isoenzymes.

## 2. Results and Discussion

### 2.1. Laccase Activity in PD and EPM Medium

To select a medium with high laccase enzyme production, the activity of laccase in PD and enzyme-producing medium (EPM) was determined. Results showed that laccase activity from EPM culture was significantly greater than that of PD culture (Figure 1), which may be due to the richer nutrients in EPM medium. The laccase activity reached maximum at 4 d (23.36 U L^−1^) in PD medium, and then declined, for the EPM culture, the activity of laccase increased with the increase of incubation time, and reached 440.92 U L^−1^ at day 8. It is interesting to note that the increase in enzyme activity in the first six days was very fast, which was less pronounced after day 6.

It has been reported that *G. lucidum* is unable to decolorize RBBR in low nitrogen medium [35], suggesting that the medium component has a great influence on decolorization of dye by laccase. Numerous studies have been undertaken to optimize culture conditions for obtaining high-yield laccase production. By optimizing the composition of the culture medium, laccase activities 12 times greater than controls have been achieved, reaching approximately 240 U mL^−1^ [36]. It was reported *G. lucidum* 77,002 yielded laccase with a high activity of 141.1 U mL^−1^ within 6 d when wheat bran and peanut powder were used as nutrient sources [24]. The results of this study were consistent with this result, as EMP medium contains wheat bran, and a higher enzyme activity was obtained in this study from EMP. Culture times greater than 6 d did not lead to a much greater increase in laccase activity, therefore medium from day 6 was selected for further study.

### 2.2. Effect of Reaction Time, pH, Temperature, and Dye Concentrations on RBBR Decolorization

To explore the optimal RBBR decolorization conditions for *G. lucidum* laccase, the effects of reaction time, pH, temperature, and dye concentrations on the RBBR decolorization percentage were determined. The supernatant collected from EPM culture was used as a crude laccase. The 3 mL reaction system comprised of 1.5 mL crude laccase concentrated to 2000 U L^−1^, 1.5 mL 0.1 M acetate buffer (pH 4.5) and 50 ppm RBBR. The basic reaction condition was: 30 min, pH 4.5, 25 °C, and 50 ppm RBBR. Analysis of the following experimental conditions only changes one of these factors and the rest remain unchanged.

#### 2.2.1. The Effect of Reaction Time

The effect of reaction time on decolorization percentage of RBBR by *G. lucidum* laccase was determined within 150 min. Results (Figure 2a) revealed that RBBR decolorization was associated with an increase in reaction time, which reached a maximum amount of 60.3% at 150 min. During the initial stage of the reaction (20–90 min), the decolorization percentage increased significantly (*p* < 0.05) and tended to be more stable at later time points. The change of decolorization is very small between the time points 90 min and 150 min. At 30 min, almost 30.8% of RBBR was decolorized, accounting for approximately half of the total decolorization percentage. Therefore, the reaction time of 30 min was chosen for subsequent measurements. It is reported that approximated complete RBBR decolorization was observed by laccases in 240 h [33]. Purified Glac3 took 16 h to reach approximated 60% decolorization percentage for three dyes (AFR, RY, and MV) [37].

#### 2.2.2. The Effect of pH

Six pH values (pH 3–7) were chosen and the decolorization percentage of RBBR varied greatly at different pH values (Figure 2b). The highest decolorization percentage (30.8%) appeared at pH 4.5, followed by pH 5.0 (19.0%). At pH values greater or less than 4.5, the decolorization percentages were much less. The decolorization percentage at each pH was significantly different (*p* < 0.05), which indicates that *G. lucidum* laccase was sensitive to pH changes and weakly acidic environments (pH 4.5) were promoted *G. lucidum* laccase activity. Laccase from *Trametes pubescens* was found to be highly stable and recalcitrant under alkaline conditions (pH 7.0 to 10.0) [38], but the optimum pH of laccase from most other white-rot basidiomycetes is between 2.2–5.0 [39,40], indicating that laccases from different sources have different characteristics. It was reported that optimum pH for RBBR decolorization by *G. lucidum* was pH 4.0 [19], this may be due to differences in breed or medium.

#### 2.2.3. The Effect of Temperature

Reaction temperature has a strong influence on enzyme activity. Seven temperatures (20–70 °C) were chosen to explore RBBR decolorization by laccase (Figure 2c). Results indicate that the decolorization percentage was low at 20 °C, and was significantly greater at 25 °C (30.8%) (*p* < 0.05). With the further increase in temperature (25–40 °C), the change in decolorization percentage was not pronounced. When the temperature exceeds 50 °C, the decolorization percentage decline significantly (*p* < 0.05), especially under 70 °C, the decolorization rate is only 5.3%, indicating temperature over 50 °C is not suitable for laccase enzyme reaction. The optimum reaction temperatures of laccases from other white-rot fungi can reach between 40–70 °C [38,39,40]. According to the report, RBBR decolorization percentage reached the maximum in 60 °C, beyond which the percentage decreased sharply [19].

#### 2.2.4. The Effect of Dye Concentration

Substrate concentration influences the activity of laccase, and excessive concentration inhibits its activity. Six concentrations of RBBR (50–400 ppm) were chosen to test their effect on decolorization. Results show that the maximum decolorization percentage (46.7%) was achieved in 200 ppm RBBR (Figure 2d). The decolorization percentage was significantly increased when 100 ppm RBBR was used (*p* < 0.05). When higher concentration (>200 ppm) of RBBR was used, decolorization percentage decreased significantly with the increase of concentration, and dropped to 24.7% at 400 ppm (*p* < 0.05). RBBR strongly inhibited activity of *G. lucidum* laccases extracted by [19], and the percentages of 17.5% and 1.1% were observed under 200 and 300 ppm RBBR. While increasing the dye concentration of three dyes (AFR, RY, and MV), decolorization percentage by purified Glac3 decreased markedly, and less 40% decolorization percentage was obtained under 100 ppm dyes [37].

To explore the optimal decolorization conditions, a three-factor and three-level experiment was designed based on results. The results of the three-factor and three-level experiment showed that the decolorization percentage varies between 26.8% to 50.3% (Table 1), and the optimal condition was Treatment No. 7: 35 °C, pH 4.0 and 200 ppm in 30 min. Therefore, our research strongly demonstrates the ability of *G. lucidum* laccases to decolorize the RBBR dye without adding any further component.

### 2.3. Effect of Mediators and Metal Ions on RBBR Decolorization

Studies have demonstrated that the presence of mediators can greatly decrease the potential energy of laccase oxidation reactions and significantly increase the oxidation efficiency [41]. There is evidence that RBBR cannot be decolorized without mediators just by pure fungal laccases [42]. Different mechanisms are also involved in the natural and synthetic mediator reactions [43,44]. Aromatic compounds and metal salts can regulate the differential expression of laccase isozyme in *G. lucidum* [45]. Three different types of copper ions are contained in the active site of laccase, and metal ions as additives can have a great impact on laccase activities. To explore the effects of metals and mediators on *G. lucidum* laccase activity, we analyzed the effects of three mediators and four metals on RBBR decolorization. Treatment No. 7 (Table 1; pH 4.0, dye concentration 200 ppm, temperature 30 °C, 30 min) was selected for the following experiment. The decolorization percentage (50.0%) indicated for this condition was used as a reference value.

Three mediators, including two natural (vanillin and syringaldehyde) mediators and one synthetic (HBT) mediator, have different effects on the decolorization of RBBR (Figure 3a). For the two natural mediators, the higher the concentration, the lower the decolorization percentage, which decrease significantly from 98.7% (0.1 mM) to 61.7% (2.0 mM) for vanillin (*p* < 0.05). This may be because the natural mediator is also a substrate for the laccase reaction. The higher the mediator concentration, the more mediators that compete with the substrate, and the laccase involved in the decolorization of dye is reduced, resulting in a decrease in the degree of decolorization. Compared with vanillin, the decolorization effect under syringaldehyde was less obvious. At low concentration ranges (0.1–1 mM), the decolorization percentage did not vary significantly (55.7%–46.6%), but with further increases in syringaldehyde concentration (1.0–2.0 mM), the decolorization varied greatly, decreasing significantly from 46.6% (1.0 mM) to 16.3% (2.0 mM) (*p* < 0.05). Unlike the two natural mediators, in general, the decolorization percentage associated with the synthetic mediator HBT increased as the concentration increased, and reached maximum (74.8%) at 2.0 mM of HBT. Among the three tested mediators, vanillin showed the greatest decolorization percentages, followed by HBT. The decolorization percentage of adding 1 mM HBT was observably better than that of 0.2 and 0.5 mM [19]. The different results may be caused by different laccase and different action site and reaction efficiency with HBT. Natural mediators assist in laccase-mediated decolorization of industrial dyes, producing better results than without the mediators. For example, 90.0% and 30.0% of Reactive Black 5 and Azure B were decolorized when acetosyringone was used as a laccase mediator, and a longer reaction time (6 h) was required to attain the maximal decolorization percentages with HBT [46]. As the concentration of HBT and ABTS increased, the degradation percentages of anthracene by laccase increased [47]. Two textile industry effluents were successfully detoxified by laccase when using HBT as mediator [48]. Low-toxic acetylacetone was used as a mediator to enhance the detoxification and degradation in a laccase mediator system [49].

Four metal ions, Mn^2+^, Na^+^, Cu^2+^ and Cd^2+^, were selected to analyze the effect of metals on RBBR decolorization by laccase. Metal ions have less influence on the decolorization effect than mediator molecules. Results showed that at low concentrations (0.5 and 1.0 mM) the decolorization percentage in the presence four metals were not significantly different (*p* < 0.05) (Figure 3b). At higher concentrations (5.0 and 10.0 mM), Na^+^ and Cu^2+^ increased the decolorization while Mn^2+^ and Cd^2+^ decreased discoloration percentage. Copper treatment with appropriate concentration can promote laccase activity [50], and therefore, it is possible to further promote the decolorization effect. The decolorization percentage decreased significantly from 58.2% (0.5 mM) to 50.3% (15 mM) with the specified concentrations of Cd^2+^ (*p* < 0.05), suggesting that higher concentrations of Cd^2+^ may inhibit the activity of laccase. The presence of heavy metal ions in textile effluents may create the problem of low biodegradability, which increases the time required for biological treatment [51]. The addition of Cu^2+^, Fe^2+^, Mn^2+^ and Cd^2+^ increased laccase activity in vitro, especially Cu^2+^ (3.86-fold), and synergistic stimulation of metal ions and aromatic compounds further enhanced this effect [52]. For example, laccase was inhibited by NaCl, but not by Na_2_SO_4_. Crude laccase completely decolorized RBBR with 1.0 mM Na_2_SO_4_ and half decolorization was demonstrated when 0.1 mM NaCl was used [33]. It was reported that the dye decolorization process was inhibited due to the presence of Hg^2+^, and dye decolorization cannot be achieved without HBT [53].

### 2.4. Laccase Isoenzyme Electrophoresis and Protein Mass Spectrometry Sequencing

To explore the expression of laccase isoenzymes in EPM medium, crude laccase from the supernatant of *G. lucidum* cultured for 6 d (relative high laccase activity from Figure 1) was used for isozyme electrophoresis analysis. The BN-PAGE was stained using Coomassie Brilliant Blue (Figure 4, lane 1) and substrate ABTS (Figure 4, lane 2). Only a wide single band can be seen on the gel, which may be due to the small difference in molecular weight of *G. lucidum* laccases. The corresponding band from the gels stained by Coomassie Brilliant Blue was excised for use in mass spectrometry (MS) sequencing, and the result showed that the main protein contained in crude laccase was Glac1 (GenBank: AHA83584.1).

The regulation of laccase isoenzymes can due to either the expression of different genes or modifications after the transcription [27]. It was reported that after 1 h incubation, one *G. lucidum* laccase isozyme was observed in CBB-R 250, RB-5, and RBBR stained gels from nativePAGE results [19]. Four *G. lucidum* laccase isoforms obtained from potato dextrose agar (PDA) medium were observed by nativePAGE, including Glac3 with a molecular mass of 38.3 kDa [37]. This may be due to different components of the medium that lead to different laccase isozyme expression profiles. The multimeric nature of laccase isozymes in *G. lucidum* was evident from SDS-PAGE results, and the molecular mass of isozymes was between 40–66 kDa [45]. On day 7, three and two isoenzymes were observed in *Pleurotus ostreatus* laccases extracts obtained from basal medium and RBBR medium respectively by nativePAGE [27].

### 2.5. Transcriptional Analysis of Laccase Genes

High laccase activity was obtained in EMP medium, and transcriptional expression profiles for laccase genes were analyzed by RT-PCR under this culture condition. Results revealed that some laccase genes have higher transcriptional expression levels (e.g., *Glac1*, *Glac5*, etc.) relative to others, whereas some genes showed lower expression (e.g., *Glac2*, *Glac10*, etc.) or were not expressed (e.g., *Glac6-8*, etc.) (Figure 5). Consistent with the above MS result, the relative expression of *Glac1* was much greater than other laccase genes; this was 10 expression level units greater that of the second expressed genes (*Glac5*). Results indicate that *Glac1* may be the main metabolite from *G. lucidum* mycelium in EPM medium. The important roles of *Glac1* in *G. lucidum* mycelium has been reported in many previous studies [54,55]. *Lcc1* from *G. tsugae* was reported to play key roles in growth and development of *G. tsugae*, such as pigmentation and stipe elongation [56]. *Lac1*, which was cloned from white-rot fungus *Cerrena sp.*, was expressed heterogeneously in yeast [57].

There are two types of fungal laccase: constitutive laccase and inducible laccase. Most of the laccase found so far belong to inducible laccase, and their gene expression is prone to change with the environmental signal [58]. Many factors can regulate laccase genes transcription such as various aromatic compounds related to lignin derivatives, metal ions, nitrogen sources, and carbon sources, etc., which lead different transcriptional profiles not only among different species, but also within the same strain [59,60]. To test whether *Glac1* plays a major role in the RBBR decolorization process, in vivo RBBR decolorization by *G. lucidum* mycelium, and transcriptional expression patterns of laccase genes were analyzed. Results revealed that the transcriptional expression level of laccase genes changed greatly when 200 ppm of RBBR was added to EPM medium after culture for 8 d. Six laccase genes with high expression levels are shown in Figure 6 and the relative expression levels of 9 other genes are shown in Appendix A.

**Except for *Glac11*:** the other five genes in Figure 6 greatly responded at the transcriptional level to RBBR in a short time (1 h). Compared to the 0 h, the fold increase in transcriptional expression level ranged from 2.14 (*Glac4*) to 6.15 times (*Glac10*). Although *Glac10* had the greatest change in transcriptional response, the background expression level of *Glac1* was very high, which was 216.94 (0 h) and 92.97 (1 h) times greater than *Glac10*. *Glac4* was the second gene with transcriptional expression level. Increased transcriptional expression levels of the five laccase genes may produce more laccase proteins that degrade the RBBR in the culture environment. As the RBBR in the environment decreases, the transcriptional expression levels of these five laccase genes gradually decreased (5 h). Compared to the 1 h, the fold decrease in transcriptional expression level ranged from 1.3 (*Glac12*) to 20.5 times (*Glac1*). As the culture time increased beyond 5h, the transcriptional expression level began to rise again, implying that after a period of adaptation, RBBR was degraded and the strain gradually returned to normal laccase secretion levels. The transcriptional expression profile of *Glac11* was opposite to that of the other five genes, with no response at the time of addition, and as the culture time increased, the transcriptional expression level increased, indicating there was a later response of this gene to RBBR. Differential expression levels of the laccase genes are associated with induction of different dyes [27]. The other 9 genes had relatively low expression levels (Appendix A), probably because they are not expressed in the mycelial stage, and involved in some other life activities of *G. lucidum*. The expression of some laccase genes is associated with different developmental stages of fungi [61].

## 3. Material and Methods

### 3.1. Microorganism and Culture Conditions

The *G. lucidum* strain, Meizhi, was obtained from the Microorganism Department of Sichuan Agricultural University, Chengdu, China. It was cultured on sterile PDA medium for 8 days at 28 °C. When plates were fully covered with the mycelia, mycelial plugs (5 mm diameter) were used as inoculum. EPM was composed of corn flour 2.0%, glucose 1.0%, bran 2.0%, soy flour 1.0%, KH_2_PO_4_ 0.3%, and pH 6.0. Three mycelial plugs were inoculated into 250 mL Erlenmeyer flasks containing 50 mL EPM or PD medium (PDA without agar) and shaking at 28 °C, 150 rpm. Samples were collected every two days (2, 4, 6, and 8 d). To test the effect of RBBR (Rhawn, Shanghai, China) on *G. lucidum* laccase, 50 ppm RBBR was added to the medium after culturing for 8 d, and samples were collected at four time intervals: 0, 1, 5, and 10 h. Experiments were performed in triplicate.

### 3.2. Enzyme Assays

Samples were centrifuged at 12,000 rpm, and supernatants were used for the laccase activity test. Laccase activity was determined by measuring changes in absorbance at 420 nm with the extinction coefficient ɛ_420_ = 36,000 M^−1^cm^−1^, using ABTS (BioDuly, Nanjing, China) as the substrate. The assay mixture contained 2900 μL of substrate (0.5 mM ABTS in 0.1 M acetate buffer, pH 4.5) and 100 μL supernatant and was incubated at 30 °C for 3 min. Activities are expressed in international units (U mL^−1^). 1 μM ABTS oxidized by laccase within 1 min is defined as an enzyme unit.

### 3.3. Dye Decolorization

The culture collected from EPM medium was centrifuged, and the supernatant was concentrated to one-half volume using an ultra-filtration centrifuge tube (30 kDa). The concentrated supernatant was diluted to an enzyme activity of 2000 U L^−1^ with 0.1 M acetate buffer (pH 4.5). The dye decolorization was measured by monitoring the decrease in absorbance at 592 nm in a UV-Vis Spectrophotometer. The reaction system was 3 mL, which contained 1.5 mL crude enzyme (2000 U L^−1^) and 1.5 mL dye solution prepared in acetate buffer (pH 4.5). Various reaction conditions, including reaction time (0, 10, 20, 30, 60, 90, and 120 min), dye concentration (100, 150, 200, 300, and 400 ppm), pH (3.0, 4.0, 5.0, 6.0, and 7.0), reaction temperature (20, 25, 30, 40, 50, 60, and 70 °C), mediators (HBT, vanillin, and syringaldehyde, purchased from Rhawn, Shanghai, China) and metal ions (Mn^2+^, Na^+^, Cu^2+^, and Cd^2+^) were chosen for analysis of the decolorization percentage. The decolorization percentage was determined in terms of percentage decrease in absorbance by using the following equation: (1)D=[100(C1−C2)]/C1 where D was the decolorization of the dye in percentage (%), C1 was the OD of initial dye system, and C2 was the OD of dye system after incubation. When the dye is completely decolorized, the OD value at 592 nm is zero, and the decolorization percentage of RBBR is 100%. Decolorization was expressed in percentages. Control samples without microorganism were separately maintained in parallel to experimental samples.

### 3.4. Laccase Isoenzyme Electrophoresis and Protein Mass Spectrometry Sequencing

Native polyacrylamide gel electrophoresis was used for the analysis of laccase isoenzyme. Same volumes of laccase samples were loaded into 2 sample wells of a 12.0% NativePAGE Bis-Tris gel (Sangon Biotech, Shanghai, China), and run at a constant voltage of 150 V. After electrophoresis, the two laccase samples on the gel were separated; one was stained with the substrate (ABTS) and the other with Coomassie Brilliant Blue. A BioDocAnalyze system (Biometra, Goettingen, Germany) was used to generate digital image of the stained gel. The sample stained with Coomassie Brilliant Blue was excised for protein MS sequencing (Sangon Biotech, Shanghai, China).

### 3.5. RNA Extraction and Real-time PCR Analysis

The total RNA of *G. lucidum* mycelium was isolated using Trizol reagent (Sangon Biotech, Shanghai, China) following the manufacturer’s instructions. An amount of 1.5 μg total RNA was synthesized to cDNA by a reverse transcription kit (Tiangen, Beijing, China), and the synthesized cDNAs served as templates in subsequent PCR reactions.

Real-time amplification reactions were run in triplicate with the iCycler iQ5 thermocycler (Bio-Rad, Hercules, CA, USA). The reaction volume contained (10 μL) 12.5 μg cDNA, 5 μL SsoFast™ EvaGreen^®^ Supermix (Bio-Rad, USA), 400 nM of each primer, and nuclease-free water to a final volume of 10 μL. The same primer sequence of laccase genes (Appendix A) from Shen’s research [62] were used in this study. Reactions were performed under the following conditions: 95 °C for 2 min, followed by 40 cycles of 95 °C for 15 s, 60 °C for 15 s, and then 72 °C for 20 s. A melting curve was created to confirm that a single product was generated by each reaction. A negative control (water) was included in each run. RT-PCR products were verified by agarose gel electrophoresis and sequencing. The *G. lucidum* constitutive gene ribosomal protein L4 (*RPL4*) was used as the internal reference gene. The relative expression levels of laccase gene were indicated as percentage to the expression of internal control gene RPL4 and the following formula was used to calculate the relative expression level of each *Glac* gene: (2)$$Y\left(\%\right)=10^{-\left(\Delta Ct/3\right)}\times 100$$ where ΔCt represents the differences in the cycle threshold value of the target *Glac* gene and the control *RPL4* products. Mean values were obtained from three biological replicates.

## 4. Conclusions

Results from this study have shown that a higher laccase production was obtained from the more nutritious EPM medium than PD medium. Reaction time, pH, temperature, and dye concentration had a strong effect on laccase-mediated degradation of RBBR, and optimal decolorization conditions were 35 °C, pH 4.0 and 200 ppm in 30 min. Moreover, mediator (vanillin and HBT) and metal ions (Na^+^ and Cu^2+^) enhanced the ability of laccase to decolorize RBBR. MS and transcriptional expression analysis results showed that *Glac1* may be the main laccase-contributing gene in the decolorization system, which is of worth for further studies.

## Figures and Tables

**Figure 1 molecules-24-03914-f001:**
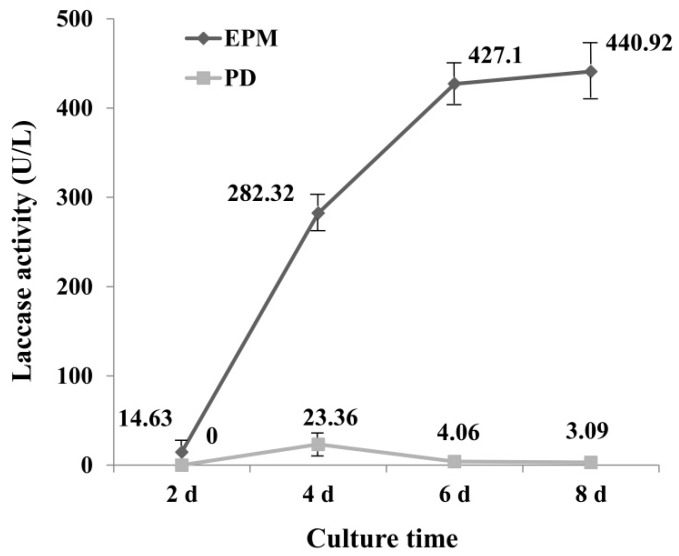
Activity of laccase under EPM and PD medium (U L^−1^). EPM = Enzyme-producing medium; PD = Potato Dextrose medium. Crude laccase was obtained from the supernatant of *G. lucidum* mycelium culture. The error bars represent the standard deviation of triplicate samples.

**Figure 2 molecules-24-03914-f002:**
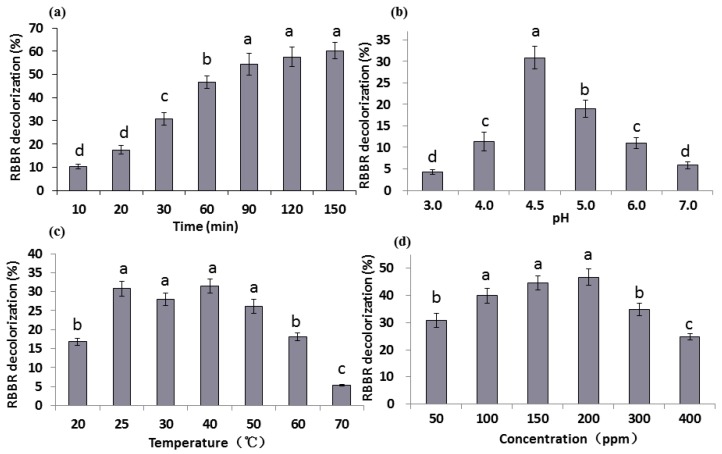
Decolorization percentage of RBBR under different conditions of: (**a**) time; (**b**) pH; (**c**) temperature; and (**d**) dye concentrations. The final laccase activity of the reaction system was 1000 U L^−1^. Decolorization percentage was determined by measuring changes in absorbance at 592 nm. The error bars represent the standard deviation of triplicate samples. Different letters (a, b, c, d) above bars indicate statistically significant differences between groups according to one-way ANOVA (*n* = 3, *p* < 0.05; refer in-text for test specifics and test statistics).

**Figure 3 molecules-24-03914-f003:**
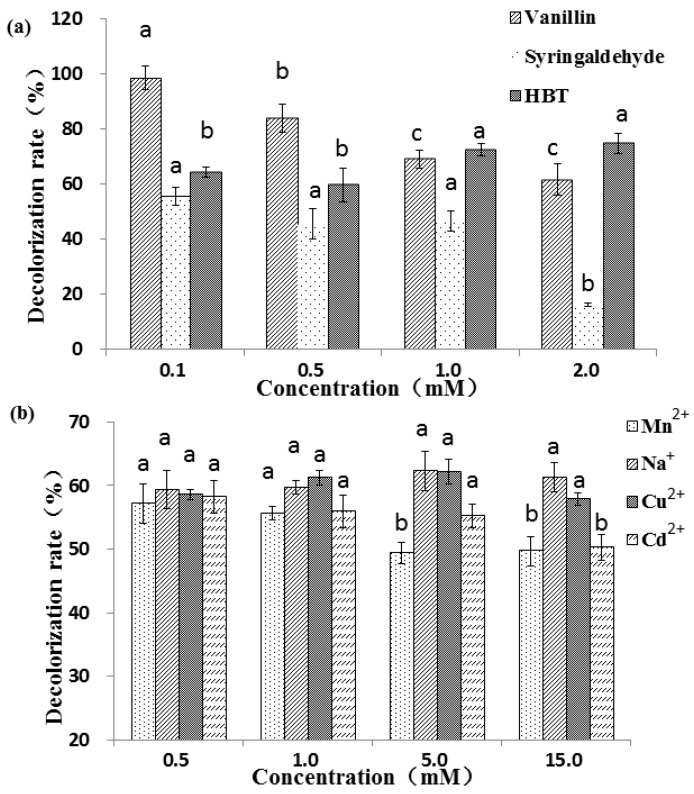
Decolorization percentage of RBBR under three mediators (**a**) and four metal ions (**b**). The final laccase activity of the reaction system was 1000 U L^−1^. Decolorization percentage was determined by measuring changes in absorbance at 592 nm. Reaction condition: pH 4.5, dye concentration 200 ppm, temperature 30 °C, and reaction time 30 min. The error bars represent the standard deviation of triplicate samples. Different letters above bars indicate statistically significant differences between groups according to one-way ANOVA (*n* = 3, *p* < 0.05; refer in-text for test specifics and test statistics).

**Figure 4 molecules-24-03914-f004:**
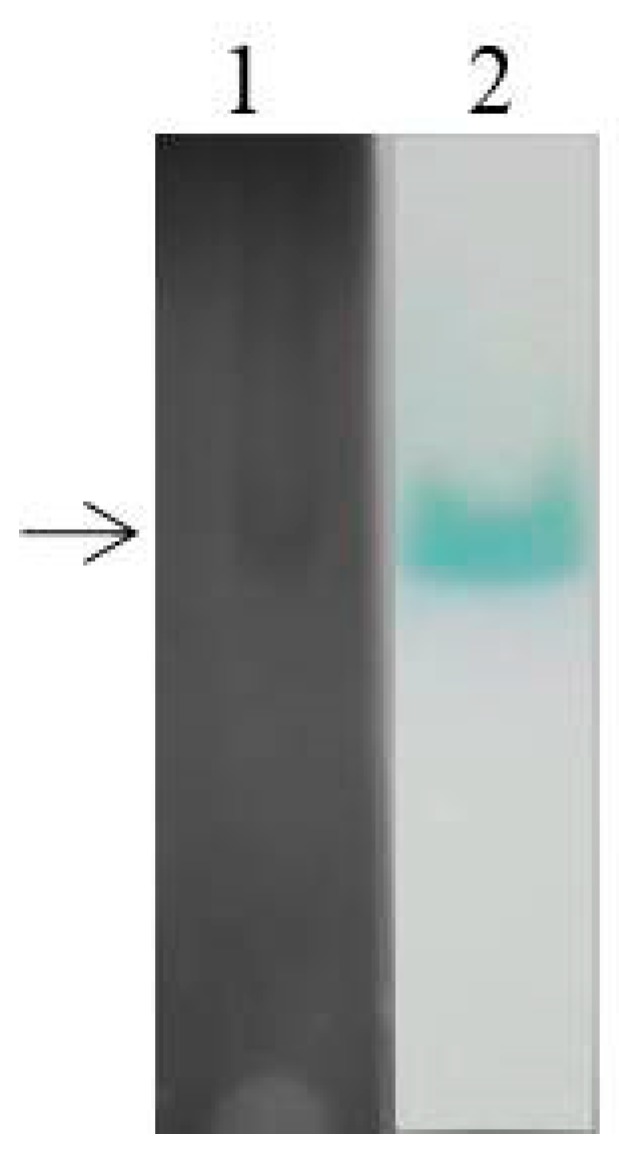
BN-PAGE analysis of laccase isozyme. The crude laccase was used for BN-PAGE. *Lane*1: Coomassie Brilliant Blue staining, *Lane* 2: ABTS staining.

**Figure 5 molecules-24-03914-f005:**
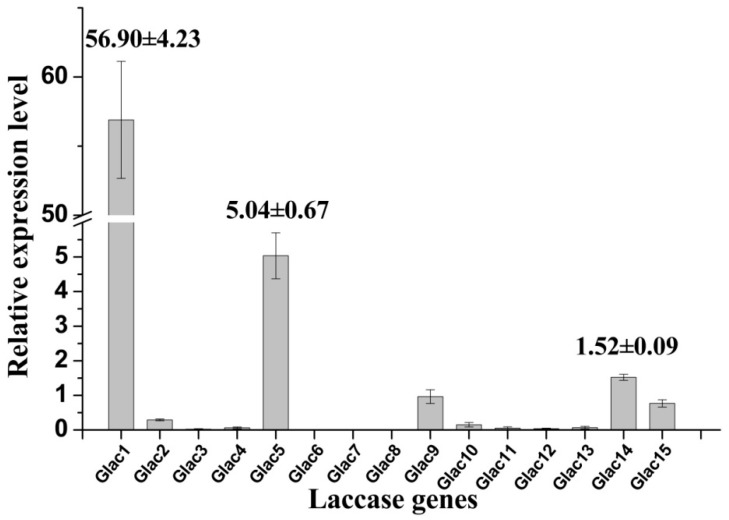
Relative expression levels of laccase gene of *G. lucidum* grown in EPM medium. The value represents the expression percentage of laccase gene accounts for the expression of *RPL4*. The error bars represent the standard deviation of triplicate samples.

**Figure 6 molecules-24-03914-f006:**
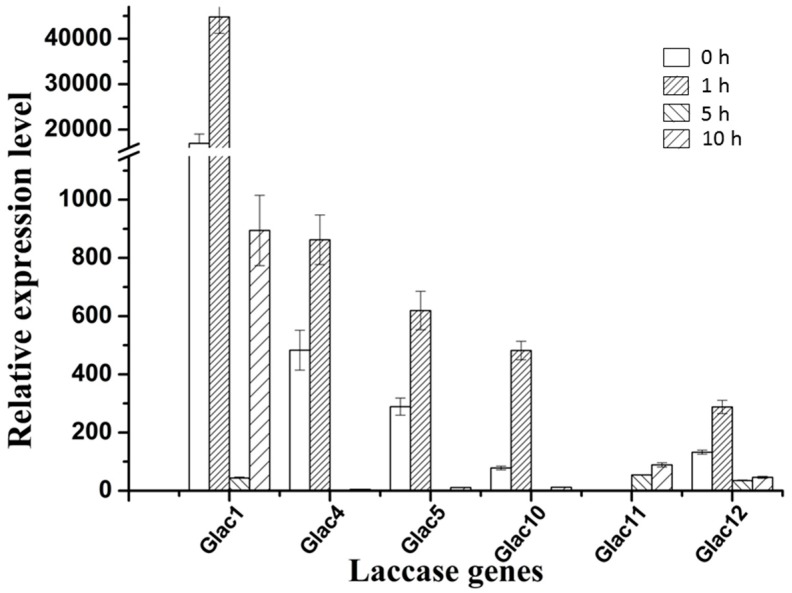
Relative expression levels of six laccase genes of *G. lucidum* with high expression levels, which were treated with 200 ppm RBBR in 4 different time periods. The value represents the expression percentage of laccase gene accounts for the expression of RPL4. The error bars represent the standard deviation of triplicate samples.

**Table 1 molecules-24-03914-t001:** Decolorization of RBBR Determined by Three-factor and Three-level Experiments.

Treat No.	Temperature (°C)	pH	Dye Concentration (ppm)	Decolorization Percentage (%)
1	25	4.0	100	39.20 ± 0.34 d
2	25	4.5	150	44.00 ± 0.85 c
3	25	5.0	200	35.41 ± 0.08 e
4	30	4.0	150	47.65 ± 0.84 b
5	30	4.5	200	49.95 ± 0.86 a
6	30	5.0	100	26.78 ± 1.00 f
**7**	**35**	**4.0**	**200**	**50.34 ± 0.30 a**
8	35	4.5	100	40.88 ± 0.58 d
9	35	5.0	150	45.49 ± 1.23 bc

The bold font is the optimal decolorization condition. Different letters (a, b, c, d, e, f) indicate statistically significant differences between groups according to one-way ANOVA (*n* = 3, *p* < 0.05; refer in-text for test specifics and test statistics).

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
