# Peer review of "Optimization of Laccase from Ganoderma lucidum Decolorizing Remazol Brilliant Blue R and Glac1 as Main Laccase-Contributing Gene"

_molecules, 2019, doi:10.3390/molecules24213914_

Round 1
Reviewer 1 Report
The authors performed extensive editing of their initial manuscript regarding the decolorization of Remazol Brilliant Blue R from Ganoderma lucidum, and the identification of the major laccase isozyme responsible for the decolorization. The changes slightly improved the quality of the manuscript, however, further changes are needed:
Lines 72-77: the added sentences do not make sense and they are unrelated with the subject of the manuscript, hence they need to be removed.
Lines 87-88: ‘…and make more substrates catalyzed by laccase.’ This sentence does not make sense. Please rephrase.
The authors did not clarify the difference between the letters a, b, c, and d in figures 2 and 3
Lines 279-287: the authors changed Figure 4, according to reviewer comments, but the text explaining the data in the figure was not changed. For example, the authors refer to lane 3 (L282) which does not exist in the revised manuscript. Please correct accordingly.
Moreover, the quality of Fig. 4, lane 1 is still very poor. I would strongly suggest to repeat the experiment and provide a better image of the gel, because in the image provided, no band is visible.
Also, authors did not provide a convincing explanation as to why the expression levels of Glac1 are so low in 5h and then they are rising again (Fig. 6), in contrast with the expression levels of all other laccases. Instead, they tried to explain these findings in lines 356-363, but this paragraph is rather confusing. To my point of view, authors support that the expression levels of Glac1 do not correlate with the addition of the dye RBBR, because ‘The expression of some laccase genes is associated with different developmental stages of fungi’ (L361-362). However, if this is the case, then the conclusion that Glac1 is implicated in RBBR degradation is not supported by the results.
Line 388: 1 mM ABTS definitely is not an international enzyme unit. This has to be changed to 1 μM.
Author Response
Response to the reviewer
Dear editor, dear Referees,
Thank you for your constructive and helpful comments concerning our manuscript. We have revised our manuscript in response to your comments. Please find a detailed point-by-point reply below. Colour coding: in “Response to Reviewers”, Referee comments are in black, our responses are in blue; in the revised version of the manuscript, all changes are marked in red.
Sincerely,
Peng Qin, on behalf of all authors
Reviewer 1
The authors performed extensive editing of their initial manuscript regarding the decolorization of Remazol Brilliant Blue R from Ganoderma lucidum, and the identification of the major laccase isozyme responsible for the decolorization. The changes slightly improved the quality of the manuscript, however, further changes are needed:
Response: Thanks for the constructive and helpful comments. According to the comments, we have carefully revised our manuscript.
Lines 72-77: the added sentences do not make sense and they are unrelated with the subject of the manuscript, hence they need to be removed.
Response: Thanks for the comments. The added sentences are about the application of Ganoderma lucidum laccase, so we summarized these sentences into one sentence as “and has been used in various fields” in the revised manuscript. (Please see page 3, line 82)
Lines 87-88: ‘…and make more substrates catalyzed by laccase.’ This sentence does not make sense. Please rephrase.
Response: Thanks for the comments. In order to briefly describe the mechanism by which the mediator affects the activity of laccase, this sentence has been restated as “Mediators can act as electron carriers between laccase and target compounds, making laccase able to oxidize not only phenolic compounds, but also non-phenolic compounds” in the revised manuscript. (Please see page 3, lines 92-94)
The authors did not clarify the difference between the letters a, b, c, and d in figures 2 and 3
Response: Thanks for the comments. We are so sorry for the unclear statement. Different letters indicate whether there is a significant difference between treatments, and a statement “Different letters above bars indicate statistically significant differences between groups according to one-way ANOVA (n=3, p < 0.05ï¼›refer in-text for test specifics and test statistics)” has been added to the legends of Fig. 2 and Fig. 3 in the revised manuscript. (Please see page 5, lines 151-153, page 8, lines 235-237)
Lines 279-287: the authors changed Figure 4, according to reviewer comments, but the text explaining the data in the figure was not changed. For example, the authors refer to lane 3 (L282) which does not exist in the revised manuscript. Please correct accordingly.
Response: Thanks for pointing out this mistake. We carefully rechecked the manuscript again and revised it.
Moreover, the quality of Fig. 4, lane 1 is still very poor. I would strongly suggest to repeat the experiment and provide a better image of the gel, because in the image provided, no band is visible.
Response: Thanks. As the reviewer suggested, we repeated the experiment for several times again. Since the color image is still not very good, so we took a black and white image with a better imaging system (BioDocAnalyze system).
Also, authors did not provide a convincing explanation as to why the expression levels of Glac1 are so low in 5h and then they are rising again (Fig. 6), in contrast with the expression levels of all other laccases. Instead, they tried to explain these findings in lines 356-363, but this paragraph is rather confusing. To my point of view, authors support that the expression levels of Glac1 do not correlate with the addition of the dye RBBR, because ‘The expression of some laccase genes is associated with different developmental stages of fungi’ (L361-362). However, if this is the case, then the conclusion that Glac1 is implicated in RBBR degradation is not supported by the results.
Response: Thanks for the comments. We are very sorry for the last unclear response. In fact, like Glac1, the expression levels of other four up-regulated genes are also low in 5 h and then they are rising again. The fold decrease in transcriptional expression level ranged from 1.3 (Glac12) to 20.5 times (Glac1) (see table below). Since the expression level of Glac1 is relatively high, the magnitude of change looks more obvious. These five genes greatly responded at the transcriptional level to the addition of RBBR in a short time, and the expression level of Glac1 was the highest. Together with the result of mass spectrometry, Glac1 may be the main contributing gene in RBBR decolorization.
Table 1 Relative expression levels of six G. lucidum laccase genes under RBBR
|
Time  |
Glac1 |
Glac4 |
Glac5 |
Glac10 |
Glac11 |
Glac12 |
|
5 h |
43.66 |
0.73 |
0.89 |
1.29 |
54.54 |
35.04 |
|
10 h |
894.49 |
4.86 |
11.40 |
12.11 |
89.16 |
46.16 |
|
10h/5h |
20.5 |
6.7 |
12.8 |
9.4 |
1.6 |
1.3 |
Laccase exists in the form of isozymes in G. lucidum, and different laccases may have different functions. Since there are several genes that are not respond to the addition of RBBR and show very low expression level, these genes may be not involved in RBBR decolorization, and may participate in other functions, such as growth and development.
Line 388: 1 mM ABTS definitely is not an international enzyme unit. This has to be changed to 1 μM.
Response: Thank you for pointing out this incorrect description, it has been corrected in the revised manuscript.
Reviewer 2 Report
The authors have addressed the comments and suggestions made by this reviewer. Thank you very much for the time considering and including all of them in the current version of the manuscript.
English language and style are fine but some minor spell check is still required
Author Response
Review 2:
The authors have addressed the comments and suggestions made by this reviewer. Thank you very much for the time considering and including all of them in the current version of the manuscript.
English language and style are fine but some minor spell check is still required
Response: Thanks for your constructive and helpful comments, and we revised the language with the help of native English speaker.
Round 2
Reviewer 1 Report
Although the authors tried to attend to all the points raised previously by the reviewers, certain issues still remain, in the revised version.
In figures 2 and 3, it is still unclear what is the difference between letters and b etc. It is clear that they refer to the statistical significance, but what is the difference between and b? In other words, why did you choose to use two different letters instead of one to present the statistical significance?
The quality of figure 4 is still unacceptable. Please remove all the gel photos, except for lanes 1 and 4
Finally, the discussion regarding the expression levels is not convincing, and rather poorly expressed. The expression levels of Glac1 rise up to 1h, then drop at 5h, and then rise again, which does not happen for other isozymes. Why is this fluctuation? This sudden drop of expression looks more like an experimental error, rather than a physiological reaction of the fungus to the RBBR levels. Is there a possible explanation (based in the relevant literature) for this? If not, perhaps this experiment needs to be repeated.
Author Response
Please see the attachment

This manuscript is a resubmission of an earlier submission. The following is a list of the peer review reports and author responses from that submission.
Round 1
Reviewer 1 Report
The authors of the present manuscript demonstrate the decolorization of Remazol Brilliant Blue R from Ganoderma lucidum as a function of laccase production by the strain, and they managed to identify the gene responsible for the major laccase enzyme produced by the strain. The manuscript is well written and organized, and the results seem convincing. A few issues that could be improved are the following:
Line 93: ‘…measuring changes in absorbance at 420 nm,..’ There is no need for so many details in the figure caption. These methods should be explained in the Materials and Methods section.
Rate is a measure of the reaction speed. The authors of this manuscript use the term ‘decolorization rate’ referring to decolorization percentage throughout the manuscript, which is not correct and can be misleading, since the factor of time is absent from the equation of line 332. ‘Decolorization rate’ must be corrected throughout the manuscript, to a more suitable term, such as ‘yield’ or ‘percentage’.
Figure 2 and 3: What is the difference between the letters a, b, c and d?
Lines 215-217: The induction of laccase genes by copper cannot explain the increase in decolorization observed in this specific assay. Please remove this sentence.
Figure 4: It is not clear what lane 2 shows. Is the excised band? If yes, it is not necessary and must be removed. Also, the band in lane 1 is not visible. Please consider repeating this experiment in order to provide a better image.
Figure 5: The error bars are missing from Glac5. On the other hand, the error bars of Glac1 are extremely high, and I am afraid that the exact value cannot be calculated by these data.
Figure 6: the expression levels of Glac1 after addition of RBBR seem a little weird. Why the expression levels are so low in 5h and then they are rising again?
L288: this sentence does not make sense, please rephrase.
Please provide accession numbers in suitable databases (such as GenBank, UniProt or NCBI Nucleotide) for all the genes and proteins mentioned in this study.
Reviewer 2 Report
The manuscript submitted by Pen Qin and coworkers is focused on potential uses of laccases from Ganoderma lucidum in decolorization of Remazol Brilliant Blue R.
The subject is of interest and fits to the scope of this journal. Despite of the presence of few interesting results related to gene transcription, the characterization of the enzymatic activity here presented is quite vague, and most of the results discusses were previously reported in other works.
Comments have been embedded trough the manuscript in order to help the authors to improve this version.
